# MnO$_x$ Supported on Hierarchical SAPO-34 for the Low-Temperature Selective Catalytic Reduction of NO with NH$_3$: Catalytic Activity and SO$_2$ Resistance

**Lusha Zhou [1], Jinkun Guan [1], Chenglong Yu [2] and Bichun Huang [1,*]**

[1] Guangzhou Higher Education Mega Centre, School of Environment and Energy, South China University of Technology, Guangzhou 510006, China; es201821035437lusa@mail.scut.edu.cn (L.Z.); esguanjk@mail.scut.edu.cn (J.G.)

[2] School of Land Resources and Environment, Jiangxi Agricultural University, Nanchang 330045, China; hjclyu@jxau.edu.cn

[*] Correspondence: cebhuang@scut.edu.cn; Tel.: +86-020-33501289

**Abstract:** The ethanol dispersion method was employed to synthesize a series of MnO$_x$/SAPO-34 catalysts using SAPO-34 with the hierarchical pore structure as the zeolite carrier, which were prepared by facile acid treatment with citric acid. Physicochemical properties of catalysts were characterized by XRD, XPS, BET, TEM, NH$_3$-TPD, SEM, FT-IR, Py-IR, H$_2$-TRP and TG/DTG. NH$_3$-SCR performances of the hierarchical MnO$_x$/SAPO-34 catalysts were evaluated at low temperatures. Results show that citric acid etching solution at a concentration of 0.1 mol/L yielded a hierarchical MnO$_x$/SAPO-34-0.1 catalyst with *ca.*15 wt.% Mn loading, exhibiting optimal catalytic activity and SO$_2$ tolerance at low temperatures. Almost 100% NO conversion and over 90% N$_2$ selectivity at 120 °C under a gas hourly space velocity (GHSV) of 40,000 h$^{-1}$ could be obtained over this sample. Furthermore, the NO conversion was still higher than 65% when 100 ppm SO$_2$ was introduced to the reaction gas for 4 h. These could be primarily attributed to the large specific surface area, high surface acidity concentration and abundant chemisorbed oxygen species provided by the hierarchical pore structure, which could also increase the mass transfer of the reaction gas. This finding suggests that the NH$_3$-SCR activity and SO$_2$ poisoning tolerance of hierarchical MnO$_x$/SAPO-34 catalysts at low temperatures can be improved by controlling the morphology of the catalysts, which might supply a rational strategy for the design and synthesis of Mn-based SCR catalysts.

**Keywords:** hierarchical SAPO-34; manganese oxides; low-temperature NH$_3$-SCR; molecular sieves; SO$_2$ resistance



## 1. Introduction

Nitrogen oxides (NO$_x$), which are released from stationary and automobile exhausts, are major atmospheric pollutants that result in many environmental issues, such as haze, acid rain and photochemical smog [1]. To date, the best developed and most efficient flue gas cleaning technology for NO$_x$ abatement from stationary sources is the selective catalytic reduction (SCR) of NO$_x$ with ammonia [2–4]. Owing to their high efficiency in eliminating NO$_x$, V$_2$O$_5$-WO$_3$/TiO$_2$ catalysts have been used commercially as an NH$_3$-SCR catalyst. However, there are some drawbacks to these commercial catalysts, including biological toxicity and low SCR performance below 300 °C, which limit their application in NH$_3$-SCR systems [1,4]. In recent years, the low-temperature NH$_3$-SCR technology, working downstream after the electrostatic precipitator, where most of the flue gas is cooled down and the SO$_2$/dusts are removed, has gained increasing attention [5,6].

Manganese-containing catalysts exhibit desirable catalytic activities at low temperatures, which received much attention in recent years [7–11]. However, Mn-based catalysts are easily deactivated when the treated flue gas contains SO$_2$. For this reason, the sulfur-poisoning mechanism of catalysts caused by SO$_2$ and SO$_3$ has been widely explored. The

active sites of catalysts were covered by ammonium bisulfate nitrate and metal sulfates deposited on the surface of catalysts, which potentially obstructed the SCR reaction at low temperatures [4,8,12]. Thus, fabricating Mn-based catalysts with both excellent catalytic activity and low $SO_2$ sensitivity at low temperatures has attracted great interest in recent years.

Researchers have suggested that the structure and morphology of catalysts in the low-temperature SCR process could enhance their $SO_2$ poisoning tolerance [13–15]. Recently, small-pore zeolites, such as SAPO-34, SSZ-13 and ZSM-5, have attracted more research owing to their exceptional catalytic activity and hydrothermal stability. Among them, SAPO-34 zeolite has a small eight-ring pore opening, a large CHA cavity and medium acidity, which has been extensively studied and used in $NH_3$-SCR reactions because of its desirable catalytic performance [16–19]. However, these microporous zeolites still have many weaknesses, including diffusion limitation of the pore/channel and reaction gas inhibition due to ammonium bisulfate (ABS) deposition, leading to catalyst deactivation in practical $NH_3$-SCR application. Hierarchical zeolites, which provide two or more levels of pore sizes, have attracted great attention. For example, Li et al. reported that 2.7 wt.% Cu/SAPO-34 with a hierarchical pore structure exhibited better $SO_2$ resistance compared to 2.7 wt.% Cu/SAPO-34 [20]. Liu et al. found that a 1.0-Cu/SAPO-34 catalyst synthesized by the one-step hydrothermal method exhibited superior catalytic activity at a temperature range of 140–430 °C [21]. Additionally, Guo et al. concluded that the decomposition rate of ABS on the SBA-15 catalyst with a larger average pore size was faster than that on conventional SBA-15 [22]. To the best of our knowledge, few studies have focused on the preparation and performance of the hierarchical $MnO_x$/SAPO-34 catalyst for low-temperature $NH_3$-SCR.

In this study, the hierarchical SAPO-34 zeolite was prepared by facile acid treatment with citric acid. The SAPO-34 zeolites etched by citric acid solutions with different concentrations were termed H-SAPO-34-x (x = 0.01, 0.1 and 0.125), in which x represents the molar concentration of the citric acid etching solution. The ethanol dispersion method was used to synthesize a series of $MnO_x$ supported on hierarchical SAPO-34 catalysts. In addition, the low-temperature $NH_3$-SCR activities and $SO_2$ tolerance of the obtained catalysts were explored. Compared with the conventional $MnO_x$/SAPO-34 catalyst, the effect of the hierarchical pore structure on SCR activity and $SO_2$ resistance at low temperatures was investigated. The results indicate that hierarchical $MnO_x$/SAPO-34 catalysts exhibited excellent low-temperature SCR activity and $SO_2$ resistance. Among these hierarchical $MnO_x$/SAPO-34 catalysts (denoted as $MnO_x$/H-SAPO-34-y, y = 0.01, 0.1 and 0.125, in which y represents the molar concentration of the citric acid etching solution), $MnO_x$/H-SAPO-34-0.1 exhibited optimal low-temperature $NH_3$-SCR performance. Characterization analyses through XRD, SEM, TEM, FT-IR, Py-IR, $NH_3$-TPD, $H_2$-TPR, TG/DTG, XPS and BET demonstrated further exploration of the impact mechanisms of the hierarchical pore structure on $SO_2$ resistance, SCR activity and stability at low temperatures.

## 2. Results and Discussion

### 2.1. Catalyst Characterization

#### 2.1.1. XRD

The ordered meso-structures and crystalline nature of catalysts were confirmed by low- and wide-angle XRD, respectively. As shown in Figure 1a, all the samples exhibited typical SAPO-34 characteristic diffraction peaks at 9.6°, 12.8°, 15.9° and 20.6° [23], and there were no impurity peaks before or after citric acid treatment. No manganese oxide characteristic diffraction peaks were observed in the XRD results for both $MnO_x$/H-SAPO-34-0.1 and $MnO_x$/SAPO-34, indicating that the Mn species were highly dispersed or persisted in an amorphous state within the catalysts. As shown in Figure 1b, there was a diffraction peak within in the range of 0.5°–1.0° for both samples, which can be assigned to the (111) reflections of an fcc structure (Fm3m), suggesting that a mesoporous structure was formed in SAPO-34 zeolites [24].

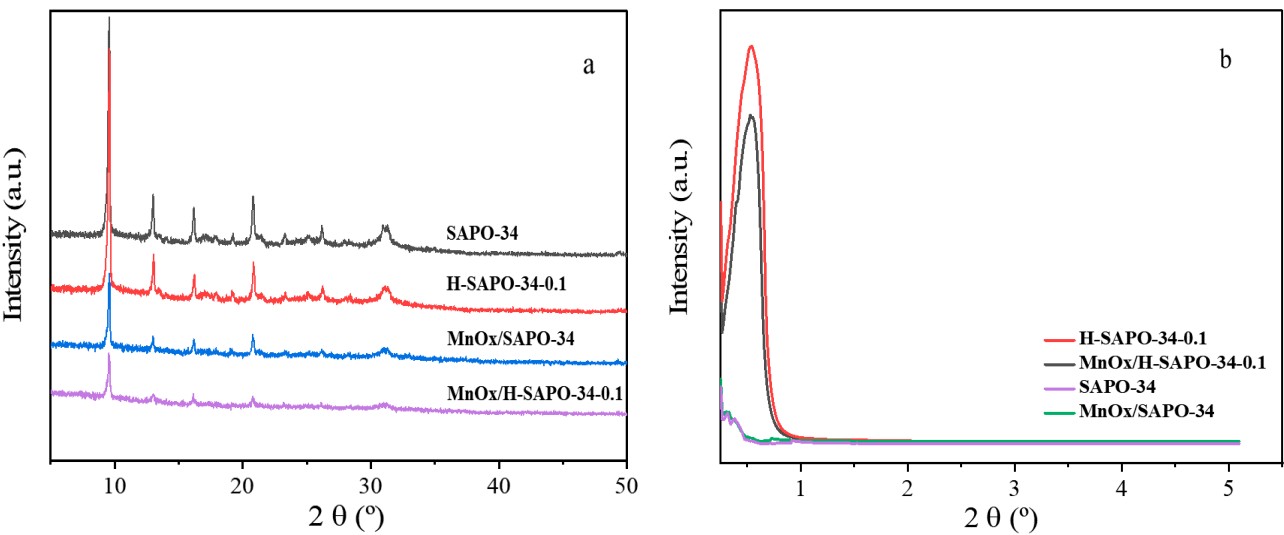

**Figure 1.** XRD patterns of SAPO-34, H-SAPO-34-0.1, (**a**) $MnO_x$/SAPO-34 and (**b**) $MnO_x$/H-SAPO-34-0.1 catalysts.

### 2.1.2. SEM and TEM

SEM was used to observe the morphological features of both the synthesized and reference zeolites (Figure 2). Cubic morphology of the characteristic CHA structure was observed in all samples. As the molar concentration of citric acid increased, the surface roughness of the SAPO-34 molecular sieve gradually increased. When the molar concentration of citric acid etching solution was 0.125 mol/L, the structure of the catalyst collapsed, which caused its external specific areas to decrease. Compared with SAPO-34, which displayed well-developed crystal faces, hierarchical SAPO-34 crystals showed a rough surface because of the surface etching with citric acid solution. As expected, the $MnO_x$/H-SAPO-34-0.1 zeolite retained its original structure well after loading manganese species, compared with the $MnO_x$/SAPO-34 sample. The fact that the $MnO_x$/H-SAPO-34-0.1 sample had much higher external specific surface areas than those of $MnO_x$/SAPO-34 prepared by conventional methods might be due to the special structure of the former. Figure 3 shows the EDS elemental mapping of the $MnO_x$/H-SAPO-34-0.1 catalyst. The elemental mapping results indicate that all elements, such as Al, Si, O, P and Mn species, were distributed homogeneously in the $MnO_x$/H-SAPO-34-0.1 catalyst. Consistent with the results of low-angle XRD, the TEM image for the $MnO_x$/H-SAPO-34-0.1 catalyst displayed in Figure 4 clearly shows that some mesopores were present within the sample. The HRTEM images (Figure 5) illustrate the lattice fringes of the $MnO_x$/H-SAPO-34-0.1 catalyst. The lattice fringe spacings of 0.311 and 0.383 nm shown in Figure 5a corresponded to the (111) plane of $MnO_2$ and the (211) plane of $Mn_3O_4$, respectively, while the lattice fringe spacings of 0.690 and 0.272 nm shown in Figure 5b corresponded to the (110) plane of $MnO_2$ and the (222) plane of $Mn_2O_3$, respectively. The most active crystal plane in the $NH_3$-SCR reaction was considered to be the (110) plane of $MnO_2$, which was highly dispersed in $MnO_x$/H-SAPO-34-0.1 [25]. This was one of the reasons for the high SCR activity of the $MnO_x$/H-SAPO-34-0.1 catalyst at low temperatures.

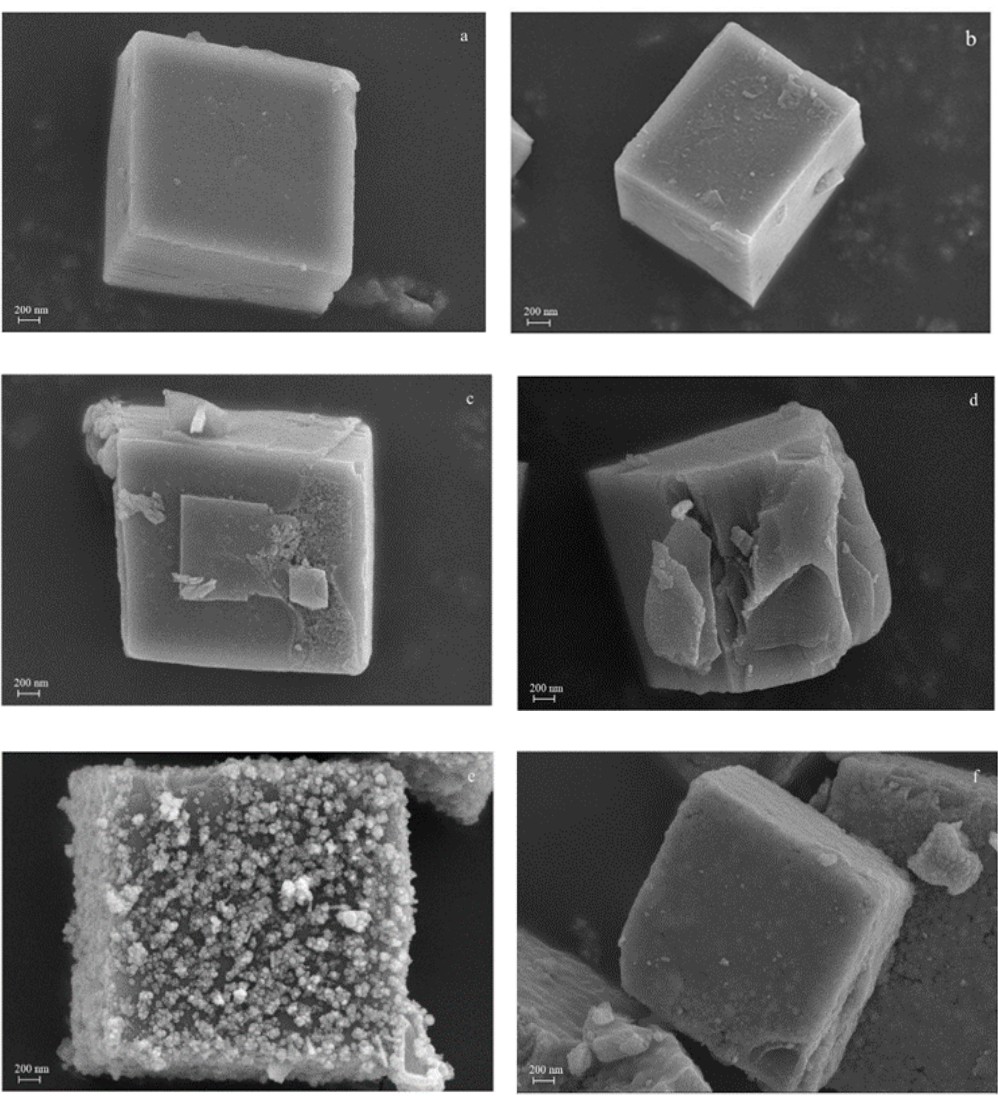

**Figure 2.** SEM patterns of SAPO-34 (**a**), H-SAPO-34-0.01 (**b**), H-SAPO-34-0.1 (**c**), H-SAPO-34-0.125 (**d**), MnO$_X$/SAPO-34 (**e**) and MnO$_X$/H-SAPO-34-0.1 (**f**).

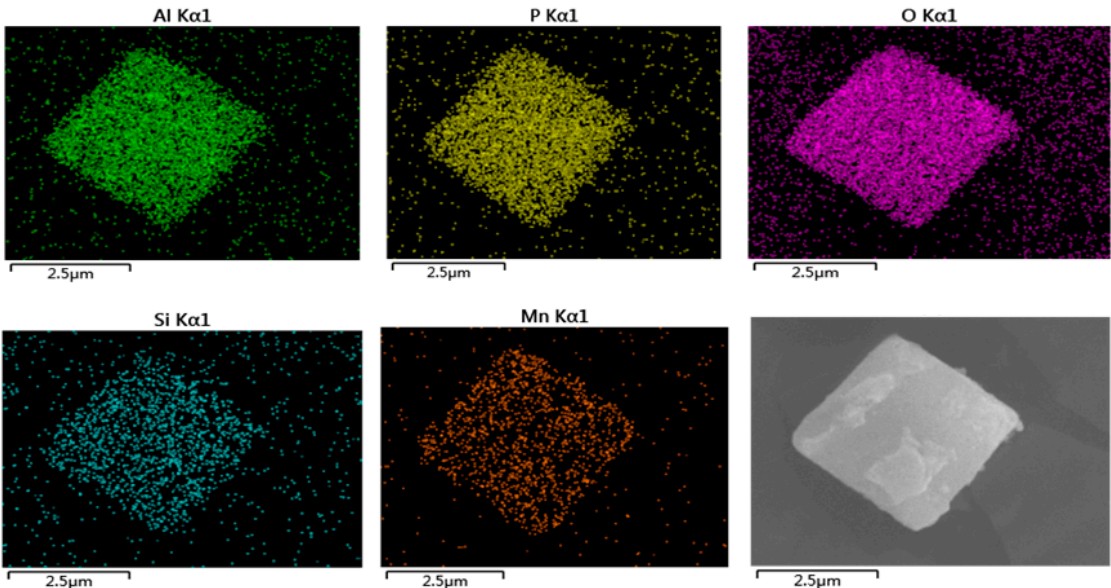

**Figure 3.** EDS mapping of MnO$_X$/H-SAPO-34-0.1 catalyst.

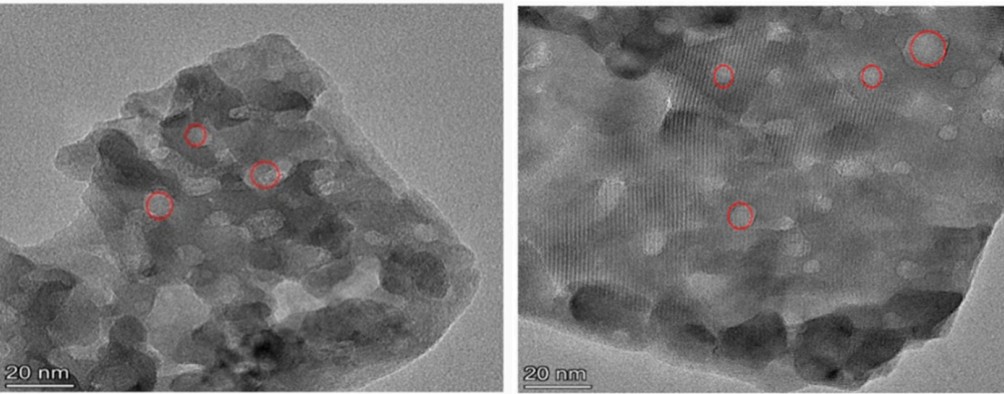

**Figure 4.** TEM patterns of $MnO_x$/H-SAPO-34-0.1 catalyst.

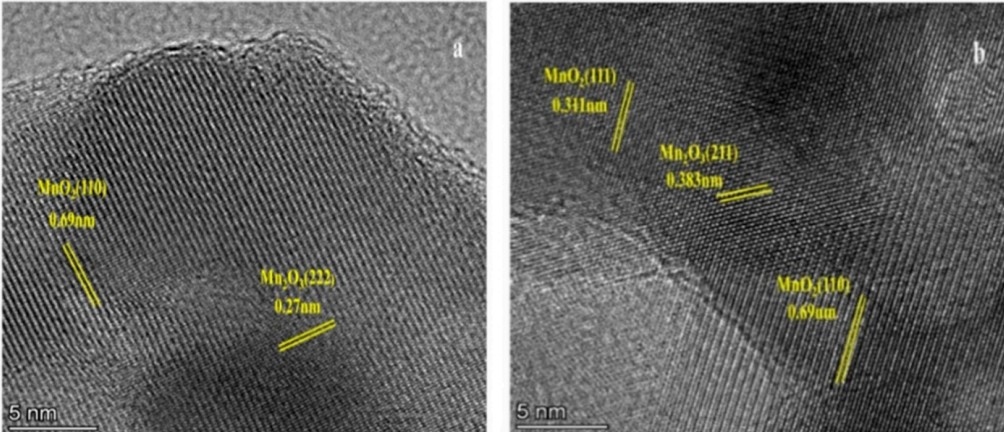

**Figure 5.** HRTEM patterns of $MnO_x$/H-SAPO-34-0.1 catalyst. (**a**) plane of $MnO_2$ and $Mn_3O_4$ (**b**) plane of $MnO_2$ and $Mn_2O_3$.

### 2.1.3. BET

The $N_2$ adsorption–desorption isotherms and pore size distribution of all samples are shown in Figure 6. As seen in Figure 6a, $MnO_x$/SAPO-34 presented typical type I adsorption–desorption isotherms in the low-pressure regions ($P/P_0 < 0.01$). $MnO_x$/H-SAPO-34-0.1 exhibited a representative characteristic type IV isotherm and a well-defined hysteresis loop at the pressure regions of $0.4 < P/P_0 < 0.9$ [26]. Hysteresis loops could be observed in the region of $0.4 < P/P_0 < 0.9$ for sample $MnO_x$/H-SAPO-34-0.1, suggesting that there were secondary larger pores in the microporous structure of the SAPO-34 crystal [27]. The pore size distribution of samples in Figure 6b,c illustrates the existence of a mesopore structure with a pore size of approximately 5 nm. The results of the structural properties of the catalysts are listed in Table 1. Compared with conventional $MnO_x$/SAPO-34, it was worth noting that the specific surface areas of $MnO_x$/H-SAPO-34-0.1 increased from 248.9 to 428.26 $m^2$/g. Meanwhile, the external specific areas of $MnO_x$/H-SAPO-34-0.1 increased from 26.61 to 37.89 $m^2$/g compared to the conventional $MnO_x$/SAPO-34 catalyst. Significantly, with respect to SAPO-34 and $MnO_x$/SAPO-34, both H-SAPO-34-0.1 and $MnO_x$/H-SAPO-34-0.1 exhibited enhancements in mesopore volume and total pore volume, with a concomitant reduction in micropore volume. Combined with the SEM and TEM results, the BET results indicate that the hierarchical $MnO_x$/SAPO-34 catalysts were successfully synthesized.

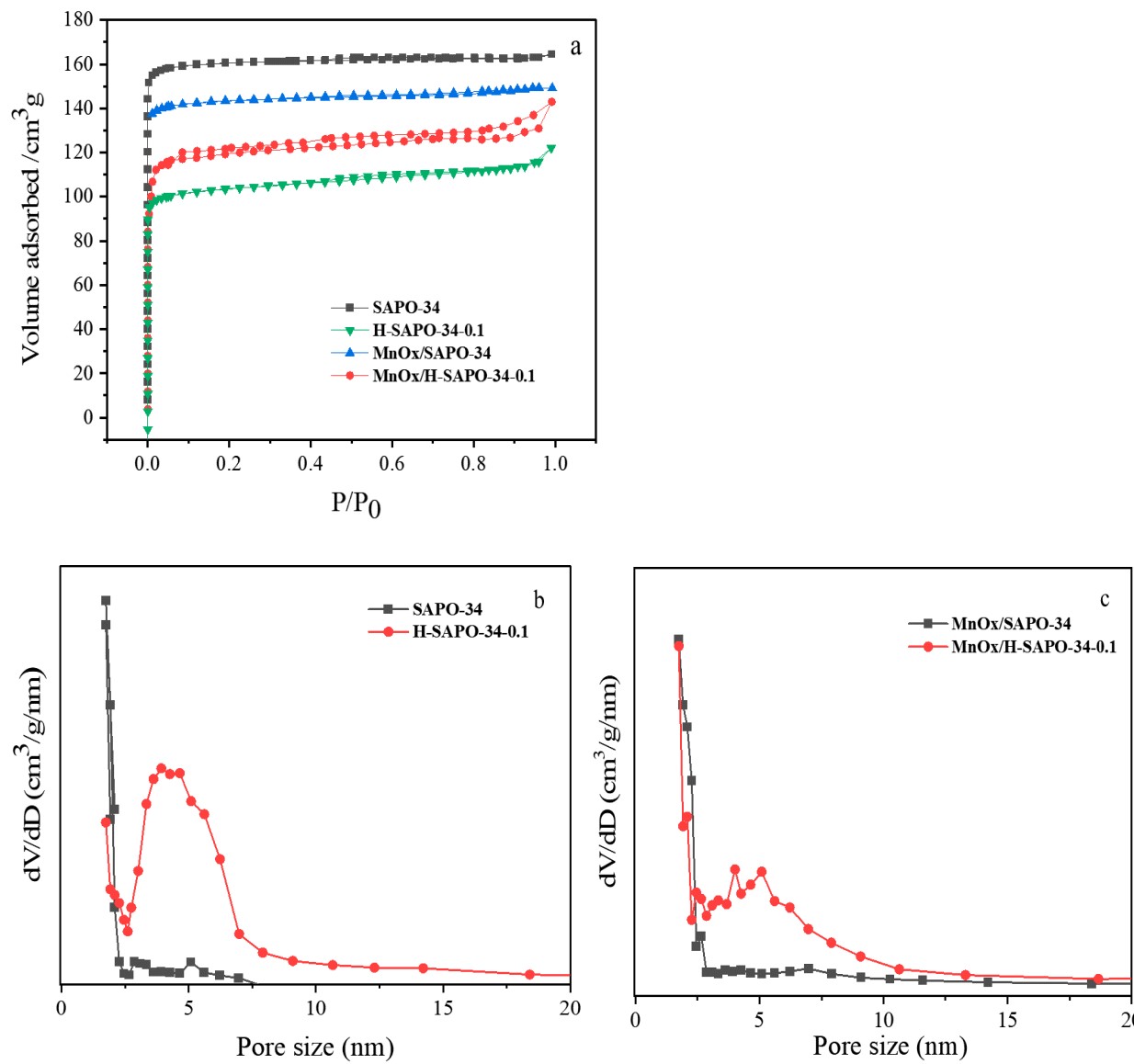

**Figure 6.** $N_2$ adsorption–desorption isotherms (**a**) and pore size distributions (**b**,**c**) of the synthesized catalysts.

**Table 1.** Structural properties of the samples.

| Samples | $S_{BET}$ [a]/ $(m^2 \cdot g^{-1})$ | $S_{mic}$ [a]/ $(m^2 \cdot g^{-1})$ | $S_{ext}$ [a]/ $(m^2 \cdot g^{-1})$ | $V_{total}$ [b]/ $(cm^3 \cdot g^{-1})$ | $V_{micro}$ [b]/ $(cm^3 \cdot g^{-1})$ | $V_{meso}$/ $(cm^3 \cdot g^{-1})$ | Ave. $D_{meso}$ [c] (nm) |
|---|---|---|---|---|---|---|---|
| SAPO-34 | 489.90 | 444.04 | 25.86 | 0.212 | 0.212 | - | - |
| H-SAPO-34-0.1 | 548.80 | 510.24 | 38.56 | 0.253 | 0.211 | 0.042 | 5.19 |
| $MnO_x$/SAPO-34 | 248.91 | 222.29 | 26.61 | 0.131 | 0.131 | - | - |
| $MnO_x$/H-SAPO-34-0.1 | 428.26 | 390.37 | 37.89 | 0.222 | 0.177 | 0.045 | 5.22 |

[a] Calculated by BET method. [b] Calculated by t-plot method. [c] Calculated by BJH method.

2.1.4. FT-IR

The chemical states of fresh and deactivated catalysts were studied by FT-IR spectroscopy and the results are illustrated in Figure 7. The sulfated $MnO_x$/SAPO-34 and $MnO_x$/H-SAPO-34-0.1 samples were marked as $MnO_x$/SAPO-34-S and $MnO_x$/H-SAPO-34-0.1-S, respectively. All of the catalysts displayed vibration absorptions centering at wave numbers of 1042 cm$^{-1}$. This was mainly due to the asymmetric stretching of the Si-O-Si bond. The bands at around 1640 and 3314 cm$^{-1}$ were caused by the stretching vibrations

of H-OH and O-H bonds from $H_2O$, respectively [28]. When the $MnO_x/SAPO-34$ and $MnO_x/H-SAPO-34-0.1$ were exposed to $SO_2$ atmosphere, both of them showed a spectrum roughly similar to that of the fresh catalysts. However, one new peak appeared at 1420 cm$^{-1}$, which could be assigned to $NH_4^+$ species chemisorbed on Brønsted acid sites. This implied that after the sulfur resistance test, ammonium species were formed on the surface of the catalyst. Concurrently, a new weak band appeared at 525 cm$^{-1}$, which might be assigned to the characteristic frequencies of the $SO_4^{2-}$ [29]. These findings indicate that sulfate species may form during the SCR reaction in the presence of $SO_2$ by binding to metal oxides or adsorbed $NH_3$ species. From the above results, the bands at 1420 and 525 cm$^{-1}$ were clearly visible for $MnO_x/SAPO-34$ after the sulfur resistance test. However, these significant peaks were not obvious in the spectrum of $MnO_x/H-SAPO-34-0.1$, indicating that sulfate species were only slightly deposited on this sample.

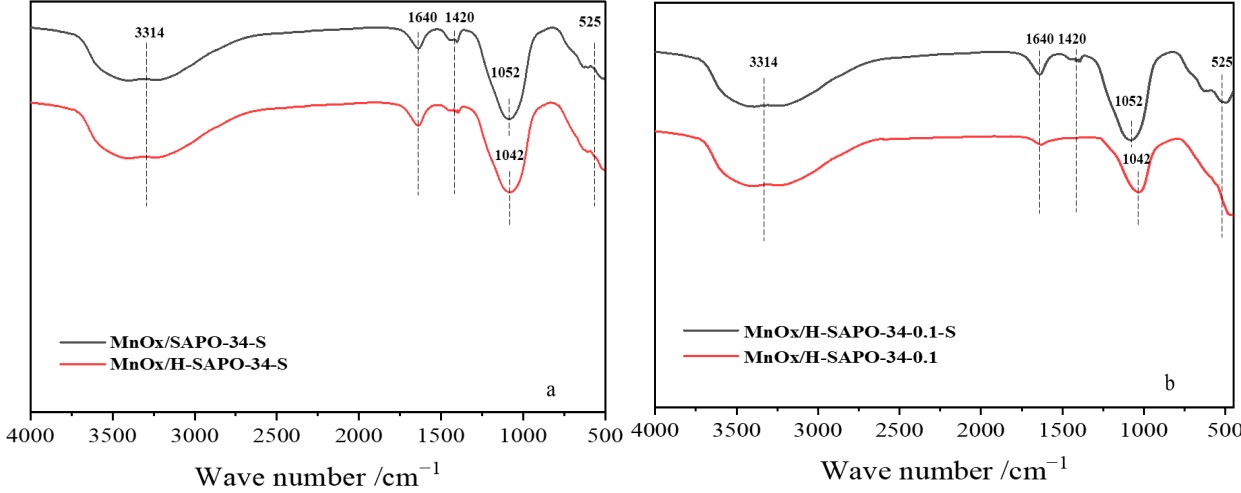

**Figure 7.** FT-IR spectrum of the fresh and deactivated $MnO_x/SAPO-34$ (**a**) and $MnO_x/H-SAPO-34-0.1$ (**b**) catalysts.

### 2.1.5. $NH_3$-TPD and Py-IR

$NH_3$-TPD experiments were conducted to probe the surface acid properties of both hierarchical and conventional $MnO_x/SAPO-34$ catalysts. As shown in Figure 8a and Table 2, the catalysts exhibited two desorption peaks in the whole temperature range. The peak from 170 to 220 °C was likely due to $NH_3$ adsorbed on physical and weak acid sites. The desorption peak at 380–530 °C was assigned to $NH_3$ adsorbed on the strong acid sites [30]. The strong acid amount of the hierarchical $MnO_x/H-SAPO-34-0.1$ was the highest among the samples, which suggests that the larger pore structure and higher surface area enhanced the concentration and acidity of strong acid sites. Therefore, the hierarchical $MnO_x/SAPO-34$ catalysts with larger specific surface areas, which could provide more strong acid sites on the surface of the catalysts for the adsorption and activation of $NH_3$, exhibited the optimal $NH_3$-SCR performance at low temperatures. Pyridine is larger than the eight-ring diameter of the CHA structure. The IR spectra of adsorbed pyridine was therefore related to the acid site in the mesopore channels [31]. The Py-IR measurement defined and established the types of acid sites in Figure 8b. Furthermore, the amounts of total and medium strong acid over the $MnO_x/SAPO-34$ catalysts were acquired with the number of desorbed pyridine molecules at 200 °C. Brønsted acid (B) and Lewis acid (L) were found to be present on both $MnO_x/SAPO-34$ and $MnO_x/H-SAPO-34-x$, corresponding to the bands at around 1535 and 1445 cm$^{-1}$, respectively. In addition, the peak at around 1490 cm$^{-1}$ was assigned to both Brønsted and Lewis acid sites. As shown in Figure 8b, the Lewis acid sites were the main acid sites of the as-obtained catalysts, and there was also a small number of Brønsted acid sites. With the increase in the concentration of citric acid etching solution, the number of Brønsted acid sites increased gradually. In addition, the

MnO$_x$/H-SAPO-34-0.1 catalyst had the highest B/L ratio (0.65) among all of the catalysts, which was favorable for the oxidation of NO to NO$_2$ for the reaction of NO +NH$_3$ + O$_2$.

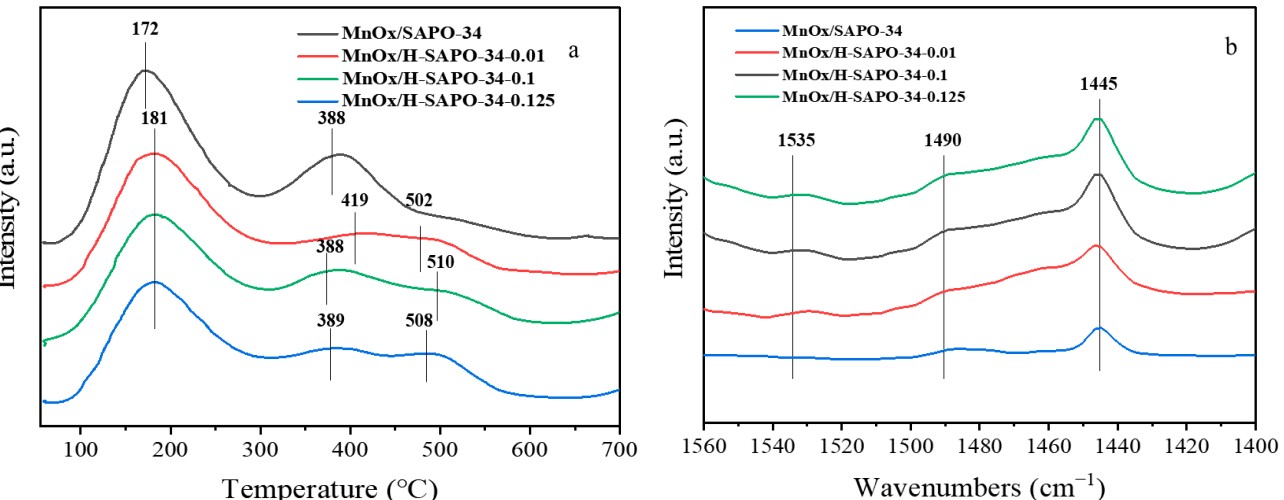

**Figure 8.** NH$_3$-TPD (**a**) and Py-IR (**b**) profiles on MnO$_x$/SAPO-34 and hierarchical MnO$_x$/SAPO-34 catalysts.

**Table 2.** Acidity data of MnO$_x$/SAPO-34 and hierarchical MnO$_x$/ SAPO-34.

| Sample | Amount of Acid Sites (mmol/g) [a] | | Amount of Acid Sites (umol/g) [b] | | B/L | Total Acidity (mmol/g) |
| --- | --- | --- | --- | --- | --- | --- |
| | Weak | Strong | Lewis Acid Sites | Brønsted Acid Sites | | |
| MnO$_x$/SAPO-34 | 0.771 | 0.316 | 70.22 | 29.78 | 0.42 | 1.098 |
| MnO$_x$/H-SAPO-34-0.01 | 0.593 | 0.327 | 65.96 | 34.04 | 0.52 | 0.899 |
| MnO$_x$/H-SAPO-34-0.1 | 0.537 | 0.377 | 60.75 | 39.25 | 0.65 | 0.914 |
| MnO$_x$/H-SAPO-34-0.125 | 0.534 | 0.336 | 61.60 | 38.4 | 0.62 | 0.873 |

[a] NH$_3$-TPD method. [b] Py-IR method.

SO$_2$ produced sulfur ammonium salt, which could cover and occupy some active sites and had a serious influence on the strong acidity of the catalysts. As seen in Figure 8, the MnO$_x$/H-SAPO-34-0.1 sample had a higher amount of strong acid sites and Brønsted acid sites than the MnO$_x$/SAPO-34 sample. This discovery indicates that the hierarchical pore structure could effectively inhibit the sulfation of active sites and provide more acid sites for the adsorption of NH$_3$ on the surface of the catalyst, which was in accordance with the results of the SO$_2$ resistance test.

### 2.1.6. H$_2$-TPR

H$_2$-TPR characterization was performed to investigate the redox properties of the catalysts, and the corresponding H$_2$-TPR profiles are shown in Figure 9. Both MnO$_x$/H-SAPO-34-0.1 and MnO$_x$/SAPO-34 catalysts presented two reduction peaks in the range of 200–800 °C. For the profile of MnO$_x$/SAPO-34, the peaks at 416 and 513 °C could be assigned to MnO$_2$/Mn$_2$O$_3$→Mn$_3$O$_4$ and Mn$_3$O$_4$→MnO, respectively [32]. Compared with the conventional MnO$_x$/SAPO-34 catalyst, it is clear that the reduction peaks shifted to a lower temperature centering at 351 and 453 °C, suggesting that the reducibility of MnO$_x$/H-SAPO-34-0.1 highly increased. Based on previous studies, the reduction peak area had a direct relationship with the consumed content of H$_2$ [33]. It can be seen that the reduction peak area of MnO$_x$/H-SAPO-34-0.1 that appeared at relatively low temperature was much larger than that of MnO$_x$/SAPO-34, suggesting that the manganese species on

the $MnO_x$/H-SAPO-34-0.1 surface is highly dispersed, which is consistent with the results of the EDS mapping [34]. In addition, the Mn species with high valence states such as $Mn^{4+}$ and $Mn^{3+}$, which were conductive to the adsorption of $NH_3$ and NO to form $Mn^{4+}$-$NH_3$ and $Mn^{3+}$-$NO_3$, could enhance the catalytic activity in the $NH_3$-SCR reaction [35]. Therefore, the $MnO_x$/H-SAPO-34-0.1 catalyst with higher reducibility exhibited optimal low-temperature performance.

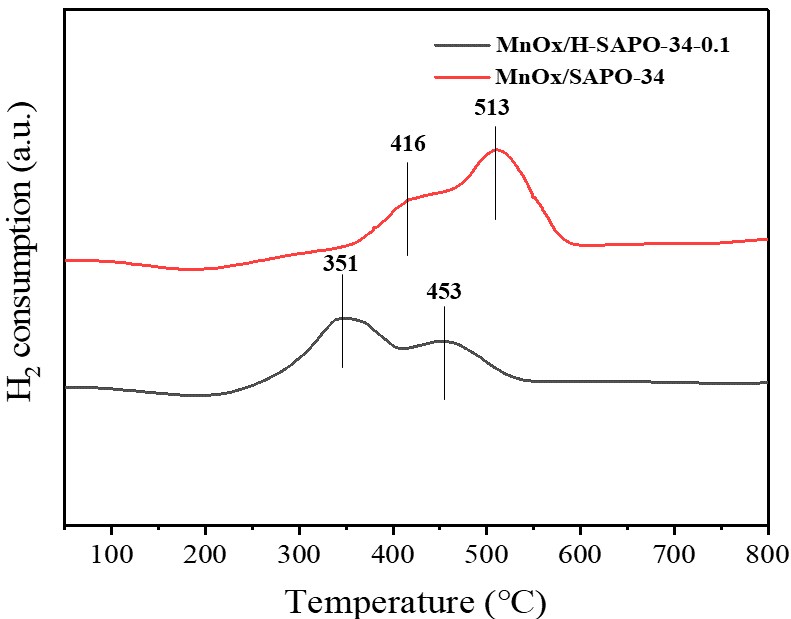

**Figure 9.** $H_2$-TPR profiles of $MnO_x$/SAPO-34 and $MnO_x$/H-SAPO-34-0.1 catalysts.

2.1.7. XPS

XPS characterization was performed to investigate the atomic distribution and composition of manganese, sulfur and oxygen ions on fresh and deactivated catalysts. The Mn $2p_{3/2}$, O 1s and S 2p photoelectron spectra are shown in Figure 10 and the assignments of the characteristic peaks in the XPS results of fresh and deactivated catalysts are listed in Table 3a. XPS-peak 4.1 software was used to perform the XPS peak fitting and background subtraction. By performing a peak fitting deconvolution, the Mn $2p_{3/2}$ spectra were divided into three peaks, $Mn^{2+}$ ($640.8 \pm 0.3$)eV, $Mn^{3+}$ ($641.8 \pm 0.2$)eV and $Mn^{4+}$ ($643.1 \pm 0.5$)eV [35,36]. O 1s spectra were separated into three characteristic peaks, which were ascribed to lattice oxygen species ($O^{2-}$ marked as $O_\alpha$) at ($529.8 \pm 0.2$)eV, chemisorbed oxygen species ($O_2^-$ or $O^-$ marked as $O_\beta$), including surface adsorbed oxygen and that of hydroxyl-like groups at ($532.1 \pm 0.2$)eV and surface hydroxyl species/adsorbed water molecules (OH or $H_2O$ marked as $O_\gamma$) at ($534.5 \pm 0.2$)eV [37,38]. In addition, the S 2p spectra can be detected in the deactivated samples (Figure 10c). The peaks at ($169.7 \pm 0.2$)eV and ($168.6 \pm 0.2$)eV are attributed to $S^{6+}/S^{4+}$ $2p_{3/2}$ and $2p_{1/2}$, respectively. Thus, it can be concluded that both $SO_4^{2-}$ and $SO_3^{2-}$ were generated on the surface of catalysts in the $NH_3$-SCR reaction with the appearance of $SO_2$ [39–44].

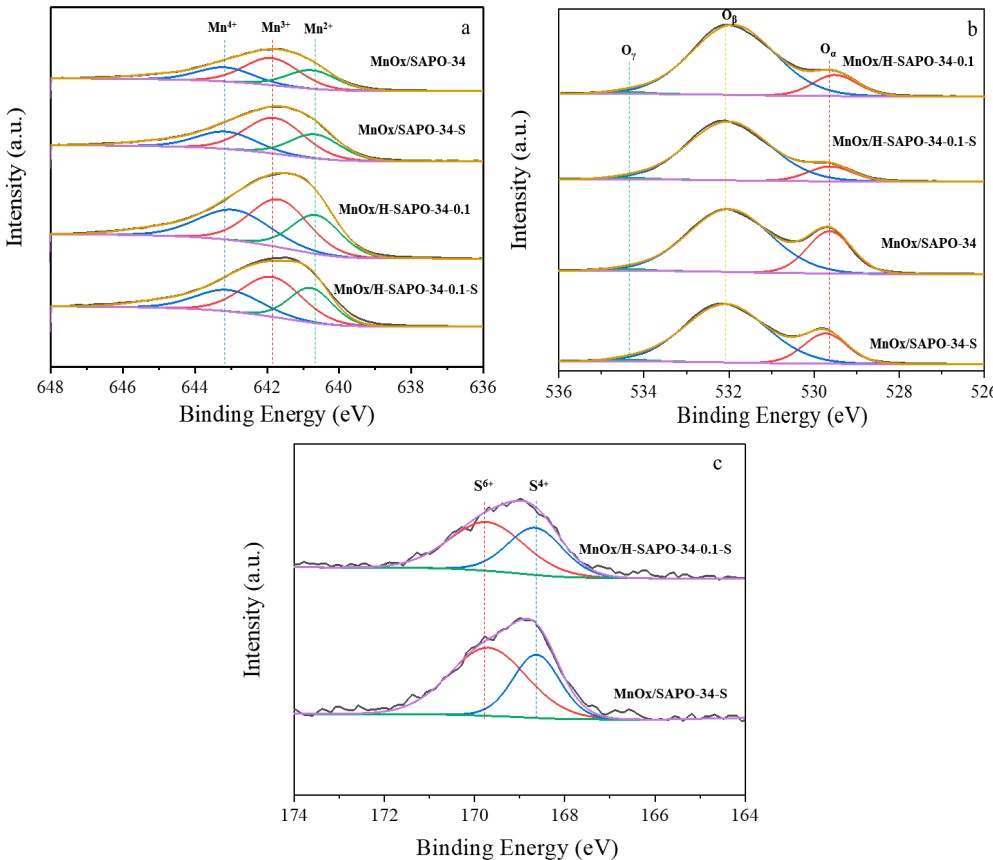

**Figure 10.** High-resolution XPS spectra of all samples: (**a**) Mn $2p_{3/2}$, (**b**) O 1 s, (**c**) S 2p.

**Table 3.** (**a**) Assignments of the characteristic peaks in XPS results of fresh and deactivated catalysts.

| Element | Binding Energy/eV | Assignment | FWHM/eV |
|---|---|---|---|
| | 529.6–530.0 | lattice oxygen species | 1.28 |
| O 1s | 531.9–532.3 | chemisorbed oxygen species | 2.39 |
| | 534.3–534.7 | surface hydroxyl species/adsorbed water molecules | 1.20 |
| | 640.5–641.1 | $Mn^{2+}$ | 1.98 |
| Mn $2p_{3/2}$ | 641.6–642.0 | $Mn^{3+}$ | 2.21 |
| | 642.6–643.6 | $Mn^{4+}$ | 1.67 |
| S 2p | 169.5–169.9 | $S^{6+}$ | 2.01 |
| | 168.4–168.8 | $S^{4+}$ | 1.22 |

(**b**) Surface compositions of all samples by XPS analysis.

| Sample | Atomic Fraction (%) | | | $X_S$ (%) | $X_O$ (%) | | | $X_{Mn}$ (%) | | | $X(Mn^{3+} + Mn^{4+})/$ $X(Mn^{3+} + Mn^{2+} +$ $Mn^{4+})$ (%) |
|---|---|---|---|---|---|---|---|---|---|---|---|
| | Mn | O | Others [a] | S | $O_\alpha$ | $O_\beta$ | $O_\gamma$ | $Mn^{2+}$ | $Mn^{3+}$ | $Mn^{4+}$ | |
| $MnO_x$/SAPO-34 | 7.87 | 59.12 | 40.88 | - | 26.26 | 72.69 | 1.05 | 28.88 | 46.46 | 24.66 | 71.12 |
| $MnO_x$/SAPO-34-S | 6.07 | 62.86 | 28.92 | 2.15 | 21.26 | 77.44 | 1.30 | 30.87 | 43.70 | 25.43 | 69.13 |
| $MnO_x$/H-SAPO-34-0.1 | 13.96 | 59.8 | 26.24 | - | 13.69 | 84.98 | 1.33 | 27.55 | 40.74 | 31.71 | 72.45 |
| $MnO_x$/H-SAPO-34-0.1-S | 11.30 | 59.51 | 27.88 | 1.31 | 11.82 | 86.17 | 2.01 | 28.05 | 48.30 | 23.65 | 71.95 |

Others [a]: Si, Al, P, O.

The results of surface compositions of all samples are shown in Table 3b. It was found that $MnO_x$/H-SAPO-34-0.1 (72.45) had a higher $(Mn^{3+} + Mn^{4+})/(Mn^{3+} + Mn^{2+} + Mn^{4+})$ ratio than that of $MnO_x$/SAPO-34 (71.12), which is consistent with the results of $H_2$-TPR. The ratios of $(Mn^{3+} + Mn^{4+})/(Mn^{3+} + Mn^{2+} + Mn^{4+})$ for $MnO_x$/H-SAPO-34-0.1 and $MnO_x$/SAPO-34 decreased from 72.45% to 71.95% and from 71.12% to 69.13% after sulfation, respectively. However, this ratio over sulfated $MnO_x$/H-SAPO-34-0.1 stayed at a high level, even higher than that over fresh $MnO_x$/SAPO-34. It has been indicated that $Mn^{4+}$ species and their redox processes promoted the catalytic cycle of $NH_3$-SCR of $NO_x$ at low temperatures [37,45]. Therefore, $MnO_x$/H-SAPO-34-0.1 exhibited excellent low-temperature activity and high tolerance to $SO_2$.

Furthermore, compared with $MnO_x$/SAPO-34 (72.69), $MnO_x$/H-SAPO-34-0.1 (84.98) had a higher ratio of $O_\beta/(O_\alpha + O_\beta)$. It has been reported that higher percentages of $O_\beta$ promoted the catalytic cycle of $NH_3$-SCR of $NO_x$ [46]. Therefore, the morphological features of $MnO_x$/H-SAPO-34-0.1 indicate that the hierarchical pore structure could produce extra surface vacancies to activate oxygen [33]. The hierarchical pore structure appeared to facilitate the transformation of NO to $NO_2$, which improved the SCR performance at low temperatures. It is worth noting that the atomic ratio of S in the deactivated $MnO_x$/SAPO-34 catalyst was slightly higher than that in the deactivated $MnO_x$/H-SAPO-34-0.1 sample, indicating that the sulfates did not accumulate in large quantities on the surface of the $MnO_x$/H-SAPO-34-0.1-S sample, which is in keeping with the results of FT-IR. It has been reported that the deposition of ABS and the sulfurating of active component Mn were the main causes of catalyst deactivation [47]. Therefore, the hierarchical pore structure can attain a dynamic balance between the formation and decomposition of ammonium sulfate, which could provide an increased surface area for the reaction process and prolong the retention of reactants on the catalyst surface. Moreover, the special structure decreased the possibility of surface active sites being taken up by $SO_2$ and prevented the formation of sulfates from blocking the active sites, leading to a high $SO_2$ resistance [48].

### 2.1.8. TG/DTG

In order to explore the $SO_2$ resistance of $MnO_x$/H-SAPO-34-0.1 and $MnO_x$/SAPO-34 catalysts, TG experiments were performed. The results are shown in Figure 11. As seen in Figure 11a, three weight losses are shown in the TG curves in the tested range. The weight loss below 200 °C was likely due to the evaporation of absorbed water, whereas the weight loss (A) between 200 and 450 °C was caused by the decomposition of ABS, and the latter weight loss (B) at 700–850 °C corresponded to the decomposition of $MnSO_4$ [44]. To investigate the possible pore size effect, a decomposition experiment was also performed on the $MnO_x$/H-SAPO-34-0.1 sample. The results are shown in Figure 11b. It was found that the TG signals of $MnO_x$/H-SAPO-34-0.1 displayed similar trends to those of $MnO_x$/SAPO-34. However, the weight loss (A) of $MnO_x$/SAPO-34 and $MnO_x$/H-SAPO-34-0.1 was measured at 4.05% and 3.39%, respectively. Smaller weight loss (A) was observed in $MnO_x$/H-SAPO-34-0.1, indicating that less ABS formed on the sample, which was consistent with the XPS results. Meanwhile, the weight loss (B) of $MnO_x$/SAPO-34 and $MnO_x$/H-SAPO-34-0.1 was measured at 2.12% and 1.73%, respectively. The results of BET and $NH_3$-TPD suggest that the $MnO_x$/H-SAPO-34-0.1 catalyst with a larger specific surface area along with abundant acid sites, which could offer a high-efficiency place to trigger the SCR reaction, resulted in improved $SO_2$ tolerance [49]. The TG/DTG analysis indicated that fewer sulfates were deposited on the surface of the $MnO_x$/H-SAPO-34-0.1 catalyst, which provided evidence for the anti-$SO_2$ capability of the hierarchical pore structure. Consistent with the results of $NH_3$-TPD and Py-IR, the effect of the hierarchical pore structure was to make a balance between effective acid sites and the formation/decomposition of ABS, thus enhancing the deNO$_x$ properties and suppressing the blocking effect of $SO_2$. Consequently, these phenomena suggest that the hierarchical pore structure promoted the diffusion of the reaction gas and minimized the sulfur poisoning of active Mn sites. This finding is

consistent with the results obtained with regard to the sulfur poisoning resistance of the $MnO_x/H$-SAPO-34-0.1 catalyst.

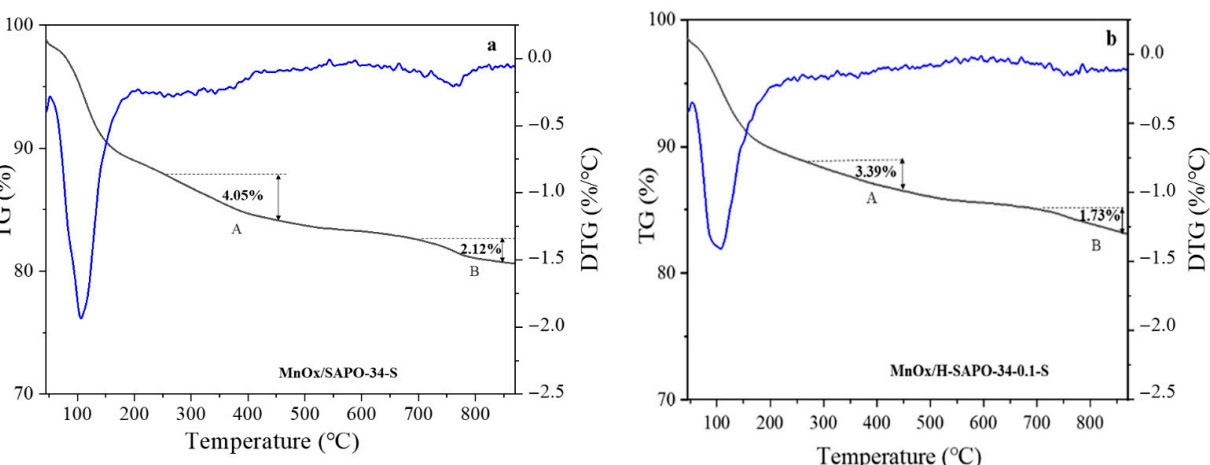

**Figure 11.** TG-DTG patterns of sulfated $MnO_x/SAPO$-34 (**a**) and $MnO_x/H$-SAPO-34-0.1 (**b**) catalysts.

## 2.2. Catalytic Performance of the Low-Temperature NH$_3$-SCR

### 2.2.1. Catalytic Activity Tests

As shown in Figure 12, performances of the hierarchical $MnO_x/SAPO$-34 and $MnO_x/$ SAPO-34 catalysts during the NH$_3$-SCR reaction were evaluated in the temperature range of 80–240 °C. To explore the influence of the hierarchical pore structure on the NH$_3$-SCR reaction, the NO conversion over $MnO_x/SAPO$-34 was measured for comparison. The $MnO_x/SAPO$-34 showed little activity until the temperature reached 120 °C, and the NO conversion gradually increased to the maximum value (about 90%) at 180 °C. At low temperatures, hierarchical $MnO_x/SAPO$-34 catalysts performed better than $MnO_x/SAPO$-34. The NO conversion with different hierarchical $MnO_x/H$-SAPO-34 catalysts as a function of the molar concentration of the citric acid etching solution in the NH$_3$-SCR reaction is shown in Figure 12. Among the hierarchical $MnO_x/SAPO$-34 catalysts, the $MnO_x/H$-SAPO-34-0.1 catalyst exhibited the optimal NO conversion of 95% with an N$_2$ selectivity over 90% in the temperature range of 80–240 °C. The superior NH$_3$-SCR performance obtained with $MnO_x/H$-SAPO-34-0.1 may due to the intercalation of mesoporous structures, which provided more channels and could substantially promote the mass transfer of reactants or products at low temperatures.

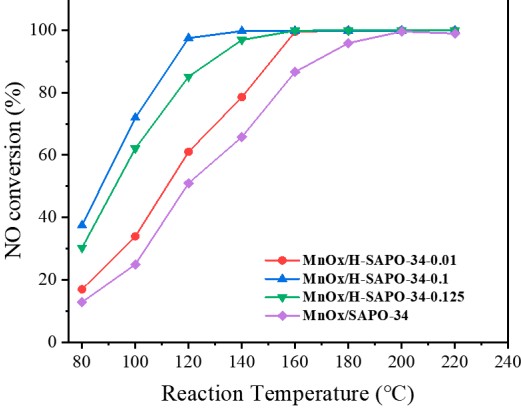

**Figure 12.** SCR activity of $MnO_x/H$-SAPO-34-0.01, $MnO_x/H$-SAPO-34-0.1, $MnO_x/H$-SAPO-34-0.125 and $MnO_x/SAPO$-34 catalysts.

2.2.2. Impact of $SO_2$ on Catalytic Activity

The effect of $SO_2$ on the catalytic performance of $MnO_x$/SAPO-34 and $MnO_x$/H-SAPO-34-0.1 catalysts was investigated at 120 °C. As shown in Figure 13, when 100 ppm $SO_2$ was added into the feed gas, a drop in NO conversion of approximately 80% occurred over $MnO_x$/SAPO-34. After removing the $SO_2$, the NO conversion did not fully recover. Nevertheless, the NO conversion of $MnO_x$/H-SAPO-34-0.1 was still maintained at 100% after the 40-min test. The $MnO_x$/SAPO-34 catalyst showed a relatively faster deactivation in a short period of time, indicating a notable difference in reaction efficiency. The SCR performance of $MnO_x$/H-SAPO-34-0.1 decreased slowly and remained above 65% for the next 200 min. Pan et al. concluded that competitive adsorption between reactant molecules and toxicants on the surface of the catalyst may promote the deactivation of the catalyst [50]. Compared to traditional $MnO_x$/SAPO-34 catalysts, the $MnO_x$/H-SAPO-34-0.1 catalyst's hierarchical pore structure was therefore found to be a key factor contributing to its high $SO_2$ poisoning resistance. The hierarchical pore structure might inhibit ammonium bisulfate aggregation and facilitate dispersion of the active phase.

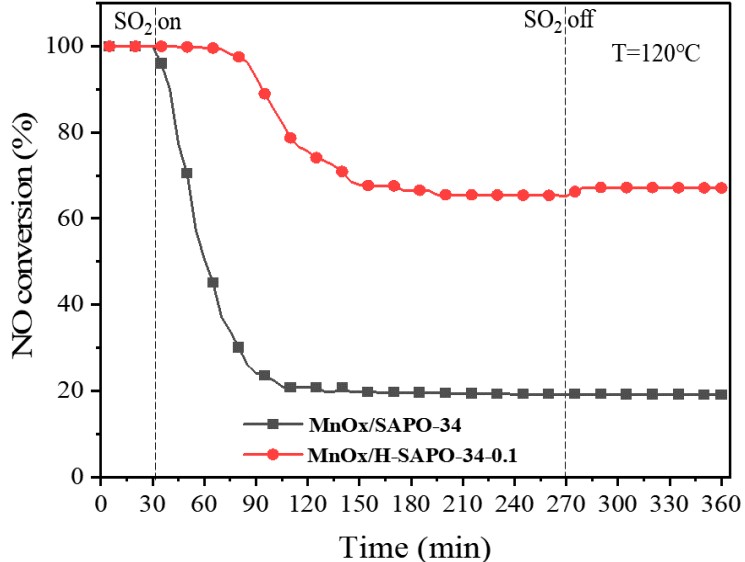

**Figure 13.** $SO_2$ tolerance of $MnO_x$/SAPO-34 and Mn/H-SAPO-34-0.1 in $NH_3$-SCR reaction.

Reaction conditions: 800 ppm $NH_3$, 800 ppm NO, 100 ppm $SO_2$ (when needed), 5 vol.% $O_2$, Ar to balance, T = 120 °C, gas hourly space velocity (GHSV) = 40,000 $h^{-1}$.

## 3. Materials and Methods

### 3.1. Catalysts Preparation

3.1.1. Preparation of Hierarchical SAPO-34

The SAPO-34 zeolite with a hierarchical pore structure (H-SAPO-34) was obtained by efficient post-synthesis via citric acid etching based on previous studies [31]. The carrier catalyst in this study was a commercial SAPO-34 with a Si/Al mass ratio of 0.27 (XFNANO Company, Nanjing, China). Typically, a certain amount of citric acid (Aladdin, Shanghai, China) was dissolved in 100 mL of ethanol and stirred for 30 min at room temperature. An amount of 3 g SAPO-34 was then added to the solution with vigorous stirring for 30 min. Then, the solution was transferred into an oil bath at 90 °C for 6 h. The product was washed with deionized water, filtered and dried overnight at 110 °C. The as-synthesized products were calcined at 550 °C for 5 h to obtain H-SAPO-34. The SAPO-34 zeolites etched in citric acid solutions with different concentrations were termed H-SAPO-34-x (x = 0.01, 0.1 and 0.125), in which x represents the molar concentration of the citric acid etching solution.

### 3.1.2. Preparation of the Catalysts

The ethanol dispersion method was used to prepare $MnO_x$/SAPO-34 catalysts with a hierarchical pore structure. Manganese nitrate (50% by weight in $H_2O$, Aladdin, Shanghai, China) was used as a precursor. The mass fraction of manganese was 15% and 2.72 mL of 50 wt.% $Mn(NO_3)_2$ was dissolved in 50 mL ethanol and stirred at ambient temperature. An amount of 3g H-SAPO-34-x was then added while being stirred. The solution was treated with ultrasound for 30 min and stirred continuously at 80 °C until the solvent evaporated completely. The products were dried at 100 °C and calcined at 400 °C for 4 h. The synthesized catalysts were denoted as $MnO_x$/H-SAPO-34-y (y = 0.01, 0.1 and 0.125, where y is the molar concentration of the citric acid etching solution). The ordinary $MnO_x$/SAPO-34 catalyst was prepared following the same method for comparison.

### 3.2. Catalysts Characterization

X-ray powder diffraction (XRD) measurement in the 2θ range of 0°–50° was obtained with the SmartLab (3KW) Japan Rigaku (Tokyo, Japan) X-ray diffractometer D8 using Cu Ka radiation (λ = 1.5418 Å). The scanning step size was $0.02°·s^{-1}$. The distribution of Mn, Si, Al, P and O species was observed using a field emission SEM in JEOL JSM-6700F (Tokyo, Japan) with X-ray energy-dispersive spectrometry (EDS). The micro-structural characterization by transmission electron microscope (TEM) images was determined by JEM-2010 (JEOL, Tokyo, Japan) with a working voltage of 200 KV. BET specific surface area and pore characterization were tested using the $N_2$ adsorption–desorption on an ASAP 2020 (Drive Norcross, GA, USA) analyzer at −195 °C. X-ray photoelectron spectroscopy (XPS) was performed through the spectrum (K-Alpha$^+$ ULTRA DLD) equipped with an Al Kα (1487 eV) radiation source.

Thermogravimetry (TG) was conducted using a STA449C-QMS403 thermal analyzer (Netisch, Germany) at a temperature range of 200–900 °C, with a heating rate of 10 °C $min^{-1}$ in 5% $O_2$/Ar. A Thermo Nicolet iS50 Spectrometer with a resolution of 4 $cm^{-1}$ was used to measure the FT-IR spectra of catalysts. $NH_3$ temperature-programmed desorption ($NH_3$-TPD) experiments were conducted on the Tp-5080 (Xianquan Industrial and Trading Co., Ltd., Tianjin, China). To explore pyridine adsorption, the sample was dehydrated at 450°C under a dynamic vacuum ($1.5 \times 10^{-3}$ Pa), followed by saturated adsorption of pyridine at room temperature. Py-IR spectra were then evacuated at 200 °C. $H_2$-TPR was conducted using a Tp-5080 (Xianqua, China) at a temperature range of 0–800 °C.

### 3.3. Catalysts Evaluation

A fixed-bed quartz flow reactor at atmospheric pressure was used to carry out SCR activity tests for the catalysts. The reaction temperature was increased from 25 to 240 °C at a rate of 5 °C/min, with an isotherm step of 20 °C. An amount of 800 mg of 40–60 mesh catalysts was used in each test. The simulated gas was composed of 800 ppm $NH_3$, 800 ppm NO, 5.0% $O_2$ and 100 ppm $SO_2$ (when needed) and balanced by Ar. All of the tests were performed with a total flow rate of 600 mL/min and a gas hourly space velocity (GHSV) of 40,000 $h^{-1}$. A $NO$-$NO_2$-$NO_x$ analyzer (Thermal Scientific, model 42i-HL, Waltham, MA, USA) was used to measure the concentrations of NO and $NO_2$. The conversion of NO was calculated by

$$C_{NO} = (1 - [NO]_{outlet}/[NO]_{inlet}) \times 100\% \tag{1}$$

where $[NO]_{inlet}$ refers to the concentration of the NO inlet gas and $[NO]_{outlet}$ refers to the concentration of the NO outlet gas.

### 4. Conclusions

In this work, citric acid treatment was used to synthesize a series of hierarchical $MnO_x$/SAPO-34 catalysts. It was found that the hierarchical pore structure improved the low-temperature activity and $SO_2$ resistance of the $MnO_x$/SAPO-34 catalyst. Among them, $MnO_x$/H-SAPO-34-0.1 presented the optimal SCR performance with more than 90% NO

conversion at 110–240 °C. Moreover, the impact of introducing $SO_2$ to $MnO_x$/H-SAPO-34-0.1 on NO conversion was lesser than that observed in the $MnO_x$/SAPO-34 catalyst. Numerous characterizations demonstrated that the hierarchical pore structure significantly increased the BET surface area and $Mn^{4+}$ percentage as well as the acid site quantity of the $MnO_x$/SAPO-34 zeolite, all of which were responsible for improving the SCR activity at low temperatures. The results of TG/DTG showed that fewer manganese sulfate species and ABS formed on the surface of the $MnO_x$/H-SAPO-34-0.1-S catalyst. The XPS results indicated that the $MnO_x$/H-SAPO-34-0.1 catalyst retained a high ratio of Mn oxides with a high valence state after sulfation. These phenomena could be ascribed to the fact that the hierarchical pore structure facilitated the decomposition of surface sulphates deposited on the catalyst during the SCR reaction, thus effectively reducing the $SO_2$ poisoning of active Mn sites. Overall, the low-temperature SCR activity and $SO_2$ tolerance of the $MnO_x$/SAPO-34 zeolite were significantly improved by the hierarchical pore structure, which could supply a rational strategy for the further design of Mn-based SCR catalysts.

**Author Contributions:** Writing, original draft preparation, L.Z.; writing, review and editing, L.Z., J.G. and C.Y.; funding acquisition, B.H.; supervision, B.H. All authors have read and agreed to the published version of the manuscript.

**Funding:** The project is financially supported by the National Natural Science Foundation of China (NSFC-51478191) and the Key Research and Development Plan of Guangdong Province (2019B110207001).

**Conflicts of Interest:** The authors declare no conflict of interest.

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
