# Peer review of "MnOx Supported on Hierarchical SAPO-34 for the Low-Temperature Selective Catalytic Reduction of NO with NH3: Catalytic Activity and SO2 Resistance"

_catalysts, doi:10.3390/catal11030314_

Round 1
Reviewer 1 Report
Review
"MnOx supported on hierarchical SAPO-34 for the low-temperature selective catalytic reduction of NO with NH3: Catalytic activity and SO2 resistance. "
In the manuscript, the authors prepared a series of MnOx supported on hierarchical SAPO with and without the treatment with citric acid. The novelty of the work is not apparent. In general, the manuscript was prepared without the necessary care and attention. English needs careful revision by the native speaker or a person with a proficient English level, as the manuscript contains plenty of lexical errors, among others. The document was submitted without proper proof-reading, and as a result, it contains many typoes that should have been corrected prior to submission. These problems make the paper difficult to read and understand. The work of the reviewer does not involve proof-reading. The abstract is not clear and should be shortened and state only the work's main findings without abbreviations if possible. There are some problems with data analysis and with some conclusions that are not supported by the data provided by the authors. The Experimental part should be checked as it contains a lot of mistakes. The paper needs significant revision before it can be considered for publication. Detailed comments can be found below.
1)Line 15 remove as the acid medium (repetition)
2)line 17 at low temperatures
3)line 21 conversion
4)line 23 remove "end."
4)line 25 "ammonia-selective catalytic (SCR) reduction catalytic activity and"- rewrite
5) most well-developed substitute for best developed
6)line 42 flue gas
7)line 51 space missing
8)line 72 In this study
9)line 79 activity and stability
10) line 82 performance of
11) As shown
12) line 292, the preparation of the catalysts was copied from another source, and the authors copied the same sentence twice. The text should have been proof-read before submission.
13) Before the results, the authors should clearly state what catalysts were prepared with the explained abbreviations. It is clear that the journal's style requires putting the experimental part at the end of the paper, but it would be useful to explain the samples either in the introduction or at the beginning of the paragraph with the results. Otherwise, it is not easy to understand the results. For example, what does x stand for in the MnOx/H-SAPO-34- x? Is it concentration of acid used?
14)line 102 replace halting with removing or stoping
15)Figure 2 is deformed. Please correct it
16) treatment with acid of the zeolites affects not only the texture but also surface chemistry. Why do the authors attribute the superior catalytic activity and SO2 resistance solely to changes in the pore structure?
17) Figure 3, how does the diffraction peak in XRD suggest mesoporous structure? It should be explained in the text.
18) line 152 the results are shown
19) what doe the authors mean by "the pore parameter"
20) line 189, which was in line with the results
21) "the hierarchical structure, which provided more acid sites on the surface of the catalyst to adsorb ammonia species" I can not agree with this statement. The acidic sites were introduced via treatment with citric acid, and the hierarchical structure was another effect of this treatment.
22) According to the authors, their samples' strong acididy came partially from produced sulfur ammonium salt. Was this salt not produced in the reference catalyst?
23) I see no reason why adsorption of SO2 should be affected by the mesoporous structure. The adsorption was more likely to be affected by the chemistry of the surface.
24) the fit of Mn species in XPS spectra should be corrected. The FWHM for all of the species in all of the spectra should be kept constant. The same applies to the fit of O1s spectra. The authors should make a table with BE and FWHM obtained from XPS analysis. Also, the type of function used for fitting should be the same. Moreover, the oxygen clearly has three, not two, different species.
25)line 270, which peaks do the authors refer to on TG graph? They should add DTG results which show the temperatures at which there was a maximum weight loss of the samples instead of Tg thermograms. It would be more relevant to the discussion that they placed in this paragraph.
26)line 276 replace "huge"
27) the results of BET – what do the authors
Author Response
Dear Reviewer:
Thank you for your such valuable comments concerning our manuscript entitled “MnOx supported on hierarchical SAPO-34 for the low-temperature selective catalytic reduction of NO with NH3: Catalytic activity and SO2 resistance” (ISSN 2073-4344). The comments are helpful for revising and improving our manuscript. We have carefully revised the manuscript according to the reviewers’ comments, and hope this will make it acceptable for publication. The attachment is our response file. Response portions are marked in red. Thanks for your attention.
Yours sincerely,
Lusha Zhou

Reviewer 2 Report
The manuscript "MnOx supported on hierarchical SAPO-34 for the low-temperature selective catalytic reduction of NO with NH3: Catalytic activity and SO2 resistance" aims at evaluating the impact of hierarchical SAPO structures on the performance of Mn catalysts for NOx processes. Despite the interest in the idea, the reviewer does not recommend publishing this manuscript in the present form.
Comments:
- The authors employ different citric acid concentrations on the different SAPO catalysts. The intention must be clearly established in the last part of the introduction. It is not clear what is to be done in the manuscript. Why they are using citric acid, why different concentrations, and what are they looking for must be indicated in the last part of the introduction.
- It is very difficult to follow the manuscript. Poor writing, missing information, and wrong work structure result in poor readability. The reader continually guessing what is the author's meaning.
- The impact of the different citric acid treatments at least on the textural properties should be clearly established prior to the catalytic activity results. This would indeed help the work's readability.
- Revise the English, the reviewer struggled reading the manuscript. For instance, the BET section is particularly messy.
- It is difficult to discern which characterization results relate to post-reacted and fresh samples. Generally, the sample nomenclature is difficult to follow.
- Add the amount of catalysts employed during the catalytic tests.
- It is missing the Mn content of the different catalysts. The evaluation of the performances of the sample requires knowing the Mn content. Assuming the nominal content is certainly not adequate.
- The differences observed in Mn4+ should be explained. The evaluation of the samples' reducibility might help to explain the different Mn species on the catalyst support.
- It is not clear the role of the porous structure of the samples in the catalysts. It might be also attributed to the acidity of the samples.
- Figures 1 and 3 captions are incomplete. What are the figures referring to is not clear. Revise them all and add some information to help the reader.
Author Response

(The authors gave the same response as above.)

Reviewer 3 Report
The manuscript by Huang and coll. describe the synthesis, characterization and evaluation of manganese oxide catalysts supported on hierarchical SAPO-34 in the low-temperature selective catalytic reduction of NO and a comparison with a non-hierarchical analog. Overall, I found this article very interesting and well written. I recommend the publication of the article following answer/correction to the comments below:
- p1 l22: 100 ppm SO2 (vs 100pm)
- It would be appreciated to see early in the text (and not only in the Materials and Methods part) the explanation of the catalyst naming, eg MnOx/H-SAPO-34-x. What H stand for? x and x? It is recommended not to use x for both MnOx and H-SAPO-34-x to avoid confusion. For instance use MnOx/H-SAPO-34-y. This would greatly help the reader.
- p3 l102: 100 ppm SO2 (not O2)
- p5 l224: Please use Mn 2p3/2 vs Mn 2P3/2.
- The authors use S2p signal to characterize the sulfur in the samples. Does the S 2p1/2 or 3/2 was used? This is not clear, while being well specified for Mn. From figure 11c, the authors claim two signals from HSO4- and SO42-; is these two peaks not rather the two spin-orbit components which are typically very closely spaced with intensity ratio of ca. 0.5?
- Fig11: Generally XPS spectra are reported from high to low BE. Fig11 a and b are reported from low to high BE, while Fig 11c from high to low. Please use the same for Figa-c, from high to low BE preferentially.
- The authors compare the weight loss (TG) of the two catalyst (p6), however there is a confusion about the first and second weight loss (l269-275), as the first weight loss is the loss of water (<150°C). It would be more clear to discuss about temperature interval to compare the weight loss: eg 250-400 and 700-850°C as mentioned by the authors (p6 l264-265). Looking at Fig12: it rather seems that the temperature interval use to determine the weight losses were 300-550°C/650-850°C for SAPO catalyst and 350-600°C/700-850°C for H-SAPO catalyst, making the comparison questionable. It is recommended to add the DTG curves for both and compare the catalysts on the same temperature interval.
Author Response

(The authors gave the same response as above.)

Round 2
Reviewer 1 Report
The manuscript is significantly improved, however, careful revision of English is still necessary. For example, some sentences do not make any sense and the style continues to be awkward. The reader should not be guessing what the authors wanted to say...
For example, in conclusions:
Moreover, introducing SO2 had a lesser impact on NO conversion
for MnOx/H-SAPO-34-0.1 compared to the MnOx/SAPO-34 catalyst.
I think the authors meant:
Moreover, the impact of introducing SO2 to MnOx/H-SAPO-34-0.1 on NO conversion was lesser than that observed in MnOx/SAPO-34 catalyst.
There is more of this kind of problems in the revised manuscript, which from my point of view should be eliminated prior to publishing.
Author Response
Dear Reviewers:
Thank you for your letter and such valuable comments concerning our manuscript entitled “MnOx supported on hierarchical SAPO-34 for the low-temperature selective catalytic reduction of NO with NH3: Catalytic activity and SO2 resistance” (ISSN 2073-4344). The comments are helpful for revising and improving our manuscript. The revised manuscript has checked by a native English-speaking colleague and use a professional English editing service. We believe that the language is now acceptable for publication.
The attachment is our revised file. Revised portions are marked in Yellow in the paper. Thanks for your attention.
Yours sincerely,
Lusha Zhou

Reviewer 2 Report
The manuscript has been re-structured, the scope of the manuscript clarified and the major uncertainties have been covered.
Author Response
Dear Reviewers:
Thank you for your letter and such valuable comments concerning our manuscript entitled “MnOx supported on hierarchical SAPO-34 for the low-temperature selective catalytic reduction of NO with NH3: Catalytic activity and SO2 resistance” (ISSN 2073-4344). The comments are helpful for us to improve the language quality of the whole article. The revised manuscript has checked by a native English-speaking colleague and use a professional English editing service. We believe that the language is now acceptable for publication.
The attachment is our revised file. Revised portions are marked in Yellow in the paper. Thanks for your attention.
Yours sincerely,
Lusha Zhou